# Microfluidic Compartmentalization Platforms for Single Cell Analysis

**DOI:** 10.3390/bios12020058

**Published:** 2022-01-21

**Authors:** Xuhao Luo, Jui-Yi Chen, Marzieh Ataei, Abraham Lee

**Affiliations:** 1Department of Biomedical Engineering, University of California, Irvine, CA 92697, USA; xuhaol@uci.edu (X.L.); juiyic1@uci.edu (J.-Y.C.); 2Department of Mechanical and Aerospace Engineering, University of California, Irvine, CA 92697, USA; mataei@uci.edu

**Keywords:** microvalves, microwells, droplets, single-cell compartmentalization, single-cell analysis

## Abstract

Many cellular analytical technologies measure only the average response from a cell population with an assumption that a clonal population is homogenous. The ensemble measurement often masks the difference among individual cells that can lead to misinterpretation. The advent of microfluidic technology has revolutionized single-cell analysis through precise manipulation of liquid and compartmentalizing single cells in small volumes (pico- to nano-liter). Due to its advantages from miniaturization, microfluidic systems offer an array of capabilities to study genomics, transcriptomics, and proteomics of a large number of individual cells. In this regard, microfluidic systems have emerged as a powerful technology to uncover cellular heterogeneity and expand the depth and breadth of single-cell analysis. This review will focus on recent developments of three microfluidic compartmentalization platforms (microvalve, microwell, and microdroplets) that target single-cell analysis spanning from proteomics to genomics. We also compare and contrast these three microfluidic platforms and discuss their respective advantages and disadvantages in single-cell analysis.

## 1. Introduction

The cell, as an indivisible unit of life, is the basic building block of all living organisms. Each individual cell provides a crucial basic structure and biological function, as well as the interplay between collective cells in response to perturbation [1,2]. Conventional cell-based assays are often used to further categorize into different types according to their morphological, functional, and other phenotypic characteristics [3,4]. This population-level cellular analysis plays an essential role in our understanding of pathogenesis and studying of cellular mechanisms. However, many conventional cell-based assays yield ensemble measurements from a population of cells, with an assumption that an averaged result is representative of a typical individual cell within a population. This generalization can often lead to misleading interpretation, given the abundance of evidence that has demonstrated cellular heterogeneity within a clonal population [5].

Substantial evidence has shown that the heterogeneity of individual cells within a population is critical for cellular function and survival in the field of immunotherapy and cancer therapy [6]. Thus, high throughput single-cell analysis is of large interest to reveal cell-cell variability by elucidating molecular mechanisms at single-cell resolution. A number of single-cell analysis methods have been developed, such as optical tweezer, patch clamp, microscopic imaging, and flow cytometry [7,8,9,10]. In particular, flow cytometry has been beneficial to single-cell analysis by detecting and enriching cells that exhibit specific phenotypic markers [11,12]. In addition, its high throughput in processing capability at a rate of thousands of cells per second makes it to be a great candidate for screening protein expression and surface markers through tagging with a fluorescent antibody or probe. However, the need for fluorescent labeling of cells limits the breadth of capability to only endpoint analysis [13]. These features, such as affinities, secretion dynamics, and cell-cell dynamic interactions, are beyond the scope of flow cytometry.

The advent of microfluidic technology offers a set of tools that enables an integrated platform for a wide range of applications from single-cell compartmentalization, manipulation, and analysis [14,15,16]. Microfluidics is the science and technology of systems that involve channels with a dimension of tens of microns to manipulate fluids at an extremely small volume ranging from femto- to nano-liter. The length scale of microfluidic systems is comparable to the size of a single cell, which is around 10 µm in diameter and roughly 1 pL in volume. The capability to operate at such miniature dimensions in microfluidic systems is attractive and provides many unique advantages over existing single-cell analysis methods. The compartmentalization of single cells in small volumes can drastically improve the signal-to-noise ratio by reducing background noise, where it would be challenging to acquire the signal of individual cells from a bulk population [17,18]. The capabilities in multiplexing and parallelization are also unique characteristics in microfluidic systems that make them suitable for single-cell analysis [19,20].

Microfluidic technologies have been evolving rapidly for a growing number of single-cell applications, which can be further divided into three main categories based on how the single cells are compartmentalized: microvalves, micro/nano-wells, and droplets. These microfluidic platforms not only offer the capability to analyze individual cells, but also increase analysis efficiency, throughput, and reduce laborious tasks that are often associated with the traditional microtiter plate method. According to Web of Science, as of 2021, the number of publications for single-cell analysis is rapidly and steadily increasing over the past two decades for all three microfluidic platforms, respectively (Figure 1). Particularly, droplet microfluidics has been an attractive method for single-cell analysis in the last decade, as indicated by its number of publications that has surpassed microvalve- and microwell-based platforms.

A large number of reviews on single cell manipulation techniques and biochemical analysis in the field of microfluidic research also reflect its rapid development and signifies its importance in driving the study of single-cell analysis. Despite many reviews being reported annually, these reviews provide different perspectives on single cell manipulation, advancement on microfluidic techniques, and various analytical methods in the field of single-cell applications [9,16,21,22]. Instead of highlighting the contribution of microfluidics in a particular technique or research field, this review will specifically focus on the recent developments of microvalve, microwell, and droplet microfluidic platforms that target single-cell compartmentalization and applications spanning from proteomics to genomics, as well as antibody discovery. We have structured this review to compare and contrast these three microfluidic platforms and discuss their respective advantages and disadvantages for various single-cell analyses.

## 2. Microvalve

Microvalves allow active control of different microfluidic processes, which enables the miniaturization and integration for a series of single-cell analysis procedures on a chip. The complicated fluid manipulation in micro-scale was made possible by the design of a pneumatic valve from Quake’s research group [23]. The system utilized multilayer soft lithography to fabricate an architecture with a flow channel lying at the bottom and a control channel perpendicularly placed on top. The layer between the channels was designed to be a thin membrane with a thickness of ~30 µm. As pressure was applied to the control channel, the membrane deformed immediately (~1 ms response time) and blocked the flow in the bottom channel, therefore the flow of fluids could be instantly and precisely turned on and off. Modifications on the design allow the fluid to be controlled by one or multiple valves and be switched to one side [23]. Further developments on the membrane enable the semi-closure of the valve to block the cells while letting the fluid pass through [24]. With the strengths of manipulating the fluids and running a sequence of reactions, microvalves facilitate the systems for isolating and analyzing single cells.

### 2.1. Genomics

A variety of treatments on the single cells can be done on-chip before the genetic studies. In some studies, single cells are pulsed with different concentrations of drugs, cytokines, and stimulants at designed intervals with the control of valves [25,26]. Tay’s group developed a system for high-throughput, multiplexed stimulation, and imaging of single cells to investigate how different dosages of TNFα affect NF-κB activity [26]. Using a similar method, the group cultured 3D tumor organoids and fed the organoids with multiple combinations of drugs at clinically relevant dosages [25]. While most operations were done on-chip in those studies, the gene amplification was done off-chip using a commercial device that performs polymerase chain reaction (PCR). On-chip PCR is strongly sought after because manually transferring the samples off-chip is a common source of contamination [27].

One example of on-chip PCR was reported by Li. et al. They developed a semi-open valve to filter, enrich, stain, and identify the circulating tumor cells (CTCs), and then performed cell lysis and multiple displacement amplification (MDA) in the same device [24]. This was done by filling the single-cell chamber with lysis reagent, neutralizing buffer, and amplification reagent in sequence with the valves. The valve not only controls the input and output of liquids but also acts as a mixer by membrane vibration. Compared to traditional tube-based MDA, the reaction volume shrank from 400 μL to 600 nL, and the reaction time was reduced from 8 to 3 h.

Another device for on-chip gene amplification is Fluidigm C1 integrated fluidic circuit (IFC) Autoprep System (Figure 2), which is by far the most commonly used device for single-cell sequencing sample preparation. The automated system can process 96 individual cells in parallel, finishing the sample preparation in 24 h. Microvalves facilitate the capturing of single cells, followed by pushing the cells into the downstream chambers for lysis, neutralization, amplification, or other treatments, depending on the target for analysis (i.e., DNA, mRNA, or exosomes). The configuration and working principle of the C1 IFC system are illustrated in the review by Prakadan et al. [28]. The system has benefited many studies [29,30,31,32,33,34] in the past few years. A total of 200 [32] to 1700 [29] single cells were analyzed in each study. The single-nucleotide variants (SNVs), insertions/deletions (INDELS), copy number variations (CNVs), and structural variants (SVs) were uncovered by whole genome amplification (WGA) and whole genome sequencing (WGS) [30] using this system. More studies are expected to be published in the coming years.

Hao et al. developed a bead-based microchip that can conduct on-chip reverse transcription quantitative real-time PCR (RT-qPCR) [27]. Single cells are trapped and lysed with the control of pneumatic valves, and the mRNA released by the cells are captured by oligo-dT-coupled- magnetic beads. A TaqMan probe was used to monitor the amplification of the targeted DNA. Besides performing RT-qPCR, this device has several other advantages over Fluidigm C1 IFC. One advantage is that it can complete a single run in 2.5 h versus the 9 h required for Fluidigm [35]. In addition, the design is simpler, with only one chamber and five valves for each working unit, compared to six chambers and seven valves in IFC [36]. However, the throughput is relatively low compared to IFC. IFC can analyze up to 96 cells at a time, whereas the microchip can only process 48 cells even after the optimization for parallel processing [37]. While the throughput is lower, the group recently made advancement on cell-tracking [38], which has not yet been reported by Fluidigm. In their study, single cells are irradiated with microbeam and then picked up by a micropipette for downstream analysis. The integration facilitates the connection between single-cell treatments and the corresponding gene expression.

The integration of valve-based microfluidics and other single-cell analysis technologies has led to great advancement in genetic profiling. Zhang et al. utilized microvalves to trap the cells, followed by droplet formation using a phase-switch device [39]. The single cells are encapsulated by hydrogel droplets, making them easily retrievable for downstream analysis. High-quality deep RNA-seq can be achieved due to the small reaction volume (200–300 pL for each droplet). Streets’ group integrated protein imaging and DNA sequencing to study the interaction between Lamin-B1 protein and lamina-associate domains (LADs) [40]. The protein-DNA interaction was measured by the DNA adenine methyltransferase identification (DamID), which identifies the methyl groups deposited near the protein-DNA contact sites at the adenine bases (^m6^A). After imaging the fluorescent ^m6^A-Tracer at the nuclear lamina, cells underwent the processes of lysis, digestion of targeted DNA site, adapter ligation, amplification, and finally off-chip sequencing (Figure 3). The integrated device allows the analysis of both spatial distribution and sequence identity, uncovering more variations of protein-DNA interactions. The group also presented another innovation, termed microfluidic cell barcoding and sequencing (μCB-seq), that pairs the imaging and sequencing information [41]. The compatibility of μCB-seq with standard microscopies makes it easy to be incorporated and implemented. It is thus a promising tool for investigating the correlation between a phenotype and the transcriptome. A tissue-dissection microfluidic device developed by Lombardo et al. demonstrated high efficiency in dissociating tissues samples into single cells [42]. While the integration of this technique has not been reported so far, it is envisioned that valve-based on-chip PCR will be incorporated with upstream single-cell preparation technologies in the near future.

### 2.2. Proteomics

Proteomes include cytoplasmic, membrane-bound, and secreted proteins. The study of proteomes in single cells involves detecting the concentration of proteins, understanding the enzymatic or functional activity, as well as investigating the signaling pathway [43]. Since protein secretion is a dynamic process and the secreted amount from a single cell is low, the limiting factors for proteomic studies are temporal resolution and sensitivity. Compared to the current macroscopic approaches, microvalves have the advantages of small sample size (pL–nL), high sensitivity, quick manipulation, high precision, and easiness of multiplexing. Therefore, it can reveal the dynamics of proteins in a more comprehensive manner. One of the earliest studies on the protein signaling pathway using microvalves is presented by Shi et al. [44]. The authors used a single-cell barcode chip (SCBC) to identify the proteins that are potentially related to receptor signaling in glioblastoma multiforme (GBM), a malignant brain tumor. SCBC consists of 120 microchambers (2 nL), each containing 11 antibody stripes to capture the proteins from a lysed cell. The study was able to map out different strengths and combinations of protein-protein interactions in a single cell. The cell signaling pathway is modulated by protein phosphorylation, which occurs in seconds to minutes. Blazek et al. designed an on-chip proximity ligation assay (PLA) to analyze the sequence of phosphorylation in the order of seconds [45]. The phosphorylation kinetics of over 500 cells was recorded with time. In summary, the valves allow for instantaneous control and tracking of transient protein levels and enable the analysis of a number of (~11–40 [44]) proteins in multiple cells.

Another major field of proteomic studies lies in antibody screening. The standard procedure employs hybridomas for antibody production. The development of microvalves facilitates the systems for the isolation and analysis of individual hybridomas, as well as the subsequent transferring and culturing of the selected cells. Indeed, systems utilizing microvalves have great strength in their ability to manipulate the fluids and run a sequence of different reactions, which pave the way for hybridoma analysis. The process includes isolating hybridomas in a chamber, capturing the proteins produced by the cells, performing immunoassays in situ, and lastly, retrieving the cells that secrete the desired antibodies. The small volume, on-chip autonomous manipulation not only expedites the screening of antibody-secreting cells but also facilitates the identification of high antibody-producing cells.

In the workflow of antibody discovery, the critical step after cell fusion is screening the hybridomas with high antibody secretion and stable proliferation. The use of a microvalve system can largely decrease the screening time and increase the accuracy. In the work of Zhang et al., the clones effectively producing anti-CD45 antibodies were identified by two rounds of screening [46]. Hybridomas were loaded into the microchamber arrays with 95% single-cell loading efficiency. The secreted anti-CD45 antibodies were captured on the surface of the chamber and were visualized by adding fluorescently labeled secondary antibodies. The fluorescent intensity at 60 min was used to quantify the antibody secretion rate, whereas the cell number per well at 48 h indicated the cell proliferation rate. The clones with the best performance were transferred to the next chamber for a second-round screening. By repeating the screening process, hybridomas with the highest secretion and proliferation rate can be identified. The valve-based, multi-round screening device increased the positive rate from 87.6% (using traditional serial dilution method on a 96-well microtiter plate) to 96.3%, reduced the detection volume from 300 µL to 0.425 nL, and shortened the overall process time from 14 to 5 days. The selected high-producing cells were transferred to a larger well for long-term culture and mass-production of antibodies. Aside from characterizing the rate of antibody secretion and cell proliferation, the specificity and affinity of the antibodies are also informative properties in profiling the antibody-secreting cells and often need to be verified before sequencing the positive clones.

In this regard, microvalve-based systems also facilitate the analysis of antibody specificity. In one study, researchers analyzed the specificity of the antibodies using microvalve-controlled chromatin immunoprecipitation (ChIP) system [47]. ChIP is usually performed by cross-linking the proteins (such as histones, transcription factors, or other DNA-associated proteins) and the DNA it binds to, shearing the protein-bound DNA strand into small fragments, and precipitating the DNA-protein complex by adding antibodies specific to the protein. Subsequently, these DNA fragments are purified and enriched by performing PCR. While ChIP is mostly used to determine the protein-DNA interactions, it is also applied to identify the specificity of antibodies by analyzing the level of gene enrichment. By using a valve-based high-throughput ChIP (HTChIP), Wu et al. compared the antibodies obtained from different sources and identified the ones with the highest sensitivity and specificity [47]. To perform the assay, antibody-coated beads were used to capture the chromatins extracted from cells in the incubation chamber. Subsequently, the beads were transferred to another chamber, washed with buffer, and finally collected for PCR analysis. This device could simultaneously scan 16 batches of antibodies using a low amount of chromatin, equivalent to only 10^4^ cells. The antibody characterization can be further improved by the development of on-chip PCR and sequencing.

Besides specificity, another area that benefits from microvalve-based systems is the characterization of antibody binding affinity, which is an essential property that gives valuable insight into the selection and administration of therapeutic antibodies. In Singhal’s study (Figure 4), functionalized beads were brought into proximity with a hybridoma to capture the secreted antibodies [48]. The subsequent washing and flushing steps were actuated by the sieve valves to immobilize the beads while allowing fluids to pass through. The beads were flushed with fluorescently labeled antigens or PBS buffer to promote antibody-antigen association or dissociation, respectively. By monitoring the fluorescent intensity of the beads over time, the association and dissociation rate constant was obtained, which provided comprehensive information on the antigen-antibody interaction. The device efficiently identified the high-affinity antibodies produced from the low-secreting cells, detecting as low as 8 × 10^4^ antibodies molecules with only 5 min of incubation time. The rapid selection of high-affinity antibodies can greatly accelerate the process of antibody discovery.

To conclude, microvalve technology has been applied to explore protein kinetics and screen the antigen-specific hybridomas due to its convenience of automation and precise fluid manipulation. It is a versatile tool that enables cell encapsulation, long-term cell culture (from days to weeks [26,49]) with various stimulation [50], and efficient cell retrieval [26]. That said, its application is still hindered by the structural complexity [28], and the throughput (<1000 cells) is limited by the number of channels (typically hundreds, or no more than a few thousand [51]) within one device. Microwell has emerged as a method with higher throughput and simplicity, thus it is more widely applied in antibody discovery.

## 3. Micro/Nano-Wells

Microwell arrays, with each well dimension ranging from 10 to 100 µm or volumes in pico-liter to nano-liter, present an approach that could improve the single-cell analysis process by increasing the screening throughput and reducing the assay time. The miniaturized well dimension enables large amounts of cells (10^3^–10^5^ cells) to be analyzed, while its tiny well volume allows protein or gene concentration to reach a detectable threshold more quickly (2–4 h) [52,53]. Microwell arrays are fabricated by making replicates from a master mold (usually a silicon wafer patterned by reactive ion etching, REI [54], or photolithography [53,55], using elastomeric materials, such as PDMS [53], PEG [56,57], and PMMA [58], to make the replicates. Cells are isolated by randomly dispersing them onto the microwells and letting them settle into the wells by gravity, followed by removing the excessive cells that are on the surface of the wells. The distribution of the number of cells per well is governed by Poisson distribution. When matching the size of the well to that of the cells or loading cells at an appropriate concentration, around 36% of wells contain single cells, and 50–80% of wells contain 1–3 cells [58,59].

Microwells have a slightly different concept than hydrodynamic traps. Hydrodynamic traps consist of chambers that have one or more symmetrical openings to let the fluid pass through [60,61]. The chambers are designed to have the dimension for holding one cell. When the chamber is occupied by a cell, the resistance of flow is high, and thus the uncaptured cells will enter other chambers. Hydrodynamic traps can reach >95% occupation efficiency when using a dense concentration of cells [62,63]. Hydrodynamic traps serve as a facile platform for cell-pairing studies [64], enabling the investigation of cell-cell interaction in various environments. While the intracellular signals, such as Ca^2+^ flux, can be analyzed, the secreted antibodies are difficult to confine, therefore challenging to be measured. Microwells, on the other hand, have the niche for analyzing antibodies secreted by single cells. Along with their ability to quickly screen and recover the cells with positive signals, microwells have gained popularity over conventional bulk methods in the field of single-cell profiling. It also has an advantage over microvalves in terms of simplicity and scalability.

### 3.1. Genomics

In both microvalve and microwell designs, the number of single cells analyzed in each device is limited by the number of microchambers. While multiplexing is doable in microvalves, the complicated multilayer fabrication restricts the large-scale expansion of the design. A microwell, on the other hand, makes it possible to process a massive number (up to a few hundred thousand) of cells in parallel, which is critical to obtaining statistically significant data for genomic profiling. Gole et al. presented the parallel MDA in single cells using a microwell array [65]. Cells were diluted to prevent having multiple cells in a well. After that, cells were loaded into the wells and lysed. The released DNA strands were denatured, neutralized, and amplified by MDA. The amplificons were extracted from the well by a glass micropipette for sequencing. The study is one of the first to demonstrate that the uniformity and coverage of whole-genome amplification are greatly improved by reducing the reaction volume to ~nL. A similar design was later applied in other studies [66,67]. Although the cross-contaminated wells can be identified and eliminated from sequencing, and no significant cross-well contamination was found, the method is still highly susceptible to contamination because the wells were not sealed.

To prevent cross-contamination between wells, Sim’s group used two methods to seal the well: a glass slide and fluorinated oil [68] (Figure 5). In addition to sealing the wells, the group captured the single-cell RNA (scRNA) onto a solid material, such as a glass slide or barcoded beads. Since the glass slide or beads are placed in the vicinity of the cell, this further ensures that the observed result is from a single cell, rather than diffused from another cell. In the first method, the group used a functionalized glass slide to cover the wells, applied vacuum to seal tightly, and then flipped the device upside down. Therefore, the RNA released from the cells was “printed” onto the glass slide. In the second method, barcoded beads were loaded into the wells filled a cell and lysis buffer, then fluorinated oil was injected into the chamber to seal the well. After incubation, scRNA was captured onto the beads for reverse transcription and amplification. The group also optimized the RNA capture efficiency by using the monodispersed beads in an appropriate size. Moreover, the automation of the system enabled efficient use of buffers while minimizing material loss [69]. The advancement provides an economical way of library preparation (USD 0.11/cell) and sequencing (USD 0.48/cell) on a large scale of cells (>2000 cells in each experiment).

An alternative approach to combat substance diffusion is Seq-Well [70]. Similar to the method reported by Sims et al. [68], a barcoded bead is settled at the bottom of a well along with a cell. The key feature of Seq-Well is the use of a semipermeable polycarbonate membrane (with 10 nm pore size) that allows the exchange of solution while confining the biological macromolecules in the well. An added advantage of the selective membrane over the glass or oil sealing is that the cell lysis can be done more efficiently, which is contributed by the supply of fresh lysis buffer from the chamber. This method has been applied to the sequencing of HEK cells, mouse 3T3 cells, and T cells [71]. Based on the Seq-Well technology, the group recently launched Seq-Well S^3^, in which a randomly primed second-strand synthesis was incorporated after the first-strand cDNA construction [72]. This significantly improved the efficiency of transcript capture and unique-gene detection. More discoveries of disease trajectories and immune cell interactions are expected to be developed from this invention.

### 3.2. Proteomics

The analysis of proteomes is rather straightforward in comparison to genomics, granting that the sequential administration of reagents is not required and that most of the effector proteins can be easily quantified by fluorescent antibodies. While microwells did not open doors for genomic research as much as microvalves did, they have been a powerful tool in proteomic studies. Microwells have been particularly useful in identifying the specific antibody-secreting cells from a heterogeneous population of cells.

Initial adaptation of microwells determined the antibody-secreting cells by detecting the intracellular Ca^2+^ signals. In the research presented by Yamamura et al., antigen-specific B cells were characterized by intracellular Ca^2+^ immobilization, which occurs as the B cell receptor engages with the antigen [58,73,74]. To visualize the intracellular Ca^2+^ immobilization, Fluo-4, the fluorescent Ca^2+^ indicator, was loaded into the cells. After being stimulated with the anti-mouse IgM antibody, the antigen-specific cells were identified and retrieved manually for single-cell RT-PCR [74].

Further improvements enable direct capture and analysis of the secreted antibodies. Love et al. presented a pioneering closed-well design, termed microengraving, to rapidly screen and selectively culture the hybridoma of interest [53]. In their work, a glass slide functionalized with either secondary antibodies or target antigens was brought in contact with the microwells (50 µm in diameter and depth), and the secreted antibodies were captured onto the glass slide (Figure 6A–C). After removing the glass slide, the fluorescently labeled antigens or secondary antibodies were dispensed onto the glass surface to identify the clones with positive reaction by visualizing the fluorescent spots. With this technique, the throughput was increased from ~384 to ~10^5^ cells/plate [53,55], whereas the time for the entire procedure, including screening, culturing, and sequencing, could be reduced from 4–6 weeks to a maximum of 2 days (including microwell preparation) [55]. As a result, this technique has fostered the studies of humoral response at the single-cell level [55,75] and offered a guideline for engineering therapeutic antibodies or vaccines with high potency. Microengraving was further expanded to analyze 42 effector proteins in a macrophage, using 15 microarrays each containing three different antibodies [76], and many immune cell functional studies [77,78] arose thanks to this technique.

Alternatively, Muraguchi et. al. presented an open-well design, in which functionalization of the capture antibody is done on the microwell itself instead of on the glass slide (Figure 6D). The presented device, immunospot array assay on a chip (ISAAC) [54,59], was fabricated by coating the secondary antibodies on the top surface of the microwells (10 µm in diameter and 15 µm in depth). The secreted antibodies were captured and then visualized by subsequently adding antigens and fluorophores. Therefore, the fluorescent signal was developed on the surface around the well that contained the antigen-specific antibody-secreting cells. This method has the benefit of locating the cells in a facile manner, as signals are directly visualized on the well itself. For microengraving, data integration is required to relate the signal on the glass back to the location on the well. While the major concept of the sealed-well and open-well systems are similar, the dynamics of antibody secretion may differ. Torres et al. characterized the transport and surface binding of the antibodies in both sealed and open-well systems [79] (Figure 6E). The study suggests that the open well is more powerful in rapid screening of cells with high antibody-secretion rates and identifying antibodies of high affinity. On the other hand, the sealed well (microengraving method) has the advantage of characterizing a broad range of secretion rates. A comprehensive analysis of these two systems, in terms of the design parameters, cell-cell interactions, and cytokine detection, can be found in the review by Park et al. [80].

The throughput of microwells can be easily scaled up by minimizing the well diameter and maximizing the number of wells in one device. However, cell retrieval from microwells is challenging, as one needs to recognize the correct well among ~10^5^ wells and gently transfer a small volume of liquid to the culture plate under a microscope. Mugraguchi’s group improved the cell-retrieval step by introducing an automated micromanipulator [59], as most microwell experiments performed with manual cell retrieval. The automated micromanipulator was able to complete the retrieval of a single cell in one minute. That said, considering the time for observing the positive clones under the microscope and a large number of cells for retrieval, further improvements are required to achieve higher throughput. In response to this limitation, Fitzgerald et al. introduced a direct clone analysis and selection technology, termed DiCAST, that took only three seconds to carry out the procedures from identifying to retrieving a cell [81]. This was done by using custom software that recognizes the positive signal, coordinates the location, and finally transfers the single cell by aspirating it from the hollow capillary to a 384-well culture plate. From the recovered hybridomas, 58% of them reached 100% confluency within 2 weeks and were confirmed to be antigen specific. Besides the efficient cell retrieval, the device scaled up the number of chambers per chip by densely packed straw-like capillaries with diameters ranging from 10 to 40 µm (80 pL–1.2 nL). With the closed-packed design, up to 15 million single-cell chambers could be placed in one chip, which was 10–1000 times higher than the previous studies [53,54,59]. Without fast and precise cell retrieval, increasing the number of chambers would only add to the burden of processing the positive cells. Efficient cell retrieval has indeed boosted the throughput of microwells.

Another technology that enables efficient cell retrieval was invented by VyCAP [82]. Their product, Puncher System, consists of 6400 microwells that are 70 µm in diameter and 360 µm in height with a 5 µm pore at the bottom. Cells are distributed in the wells by applying a small negative pressure. Antibodies from the cells diffused through the 5 µm pore, bound to the functionalized surface, and were quantified by immunofluorescence assay (Figure 7). Cell retrieval was relatively easy by using an automated system that directly punched the cells into the culture plate, with a reported survival rate of 70%. Abali et al. combined the Puncher System with a detection membrane to sort out the hybridomas producing EpCAM-antibodies [83]. The Puncher System was also reported to be capable of quantifying the secreted antibodies by surface plasmon resonance (SPR), which is a label-free real-time detection technology that senses the binding of ligands based on the change of refractive index as binding occurs [84]. Despite the novelty in this method, the operation complexity is comparable with the previous immunoassay methods since the conjugation of gold nanoparticles is required to enhance the signal. Thus, to the best of our knowledge, no later studies were found to adopt SPR for antibody detection.

Some in-situ protein analysis platforms were developed, circumventing the need for cell retrieval. Herr’s group constructed an in situ Western blot of single cells by patterning fibronectin (FN)-functionalized polyacrylamide (PA) gel into microwell structure [85] (Figure 8). Cells were isolated and incubated overnight in the microwells (50–100 μm in diameter). Once cells were lysed, an electric field was applied immediately to initiate electrophoresis. The following procedures are almost identical to the conventional Western blot, making the platform easily to apply. The device has helped investigate the effect of osmolarity on kinase phosphorylation [85] and identify the apoptotic state of cells [86].

Microwell system offers the flexibility of integrating with other systems to expand its functions. For example, integration with Illumina next-generation sequencing (NGS) enables genetic profiling of the high-affinity, antigen-reactive antibodies. The use of nanowells in combination with nanodroplets has also contributed to the liquid chromatography-mass spectrometry (LC-MS) of single-cell proteomes [87,88]. Many reviews have been published describing the integrated systems and their wide variety of applications [16,28]. Compared to microvalves, microwells greatly reduced the complexity of operation while preserving the merits of small sample volume and multiplexing. However, in both techniques, the number of cells to be processed is limited by the physical dimensions of the chip, in which multiple channels or wells need to fit in. In contrast, droplet technologies can screen unlimited numbers of cells, therefore the studies of droplet microfluidics have grown rapidly in recent years.

## 4. Droplet Microfluidics

A droplet-based microfluidic system provides another powerful alternative for single-cell analysis, yet overcoming many limitations observed in bulk systems. Briefly, two streams of immiscible phases are introduced into a microfluidic device to generate monodispersed aqueous droplets that are stabilized by a continuous oil phase supplemented with surfactant. Instead of compartmentalizing single cells in microwells, droplet microfluidics exploits immiscible phases of aqueous reagent in an oil emulsion to generate pico- to nano-liter sized cell containing droplets. Since the advent of droplet microfluidic systems, it has gained substantial interest and development of tools for high-throughput bioassays owing to the unique abilities in generating monodispersed micro-droplets at a rate of thousands per second and eliminating cross-contamination [89]. This high rate of droplet generation provides flexible scalability to perform a large number of individual assays without being limited by any physical dimensional constraint or increasing complexity of a microfluidic chip, which leads to several orders of magnitude higher throughput as opposed to the aforementioned microfluidic systems. With the biocompatible aqueous and oil phase, the encapsulated cells can maintain their viability for an extended period of time in droplets. With the strengths of encapsulating cells in droplets as micro-reactors, droplet microfluidics provides more capabilities in isolating and analyzing single cells.

### 4.1. Genomics

Droplet microfluidics has made significant progress in single-cell genomics that enables extensive studies on genomic heterogeneity in cells. The compartmentalization of a single cell in droplet offers unique opportunities to investigate genomic heterogeneity between cells and identify rare cells, which provide insights on cellular diversity, DNA mutations, and gene expressions [90].

RNA sequencing (RNA-seq) was a breakthrough in the late 2000s to measure gene expression level across a population of cells, but the ensemble measurement is insufficient for heterogeneous cell populations and understanding of stochastic nature of gene expression. The emergence of droplet microfluidics serves as a critical platform to enable RNA-seq in single-cell resolution. Single-cell RNA-sequencing (scRNA-seq) was first realized by Tang et al. to enable the study of heterogeneity of gene expression within a cell population, despite the throughput for transcriptome analysis being low [91]. With the development of chemistry for improved sensitivity, the co-encapsulation of single cell and molecular barcoded bead in a droplet can achieve higher throughput that led to several commercialized platforms, including DropSeq, inDrop, 10x Genomics, and Mission Bio Tapestri systems [92,93]. To exploit the advantage of isolating cells and molecular barcoded beads in droplets, Macosko et al. developed DropSeq platform that enables genome-wide expression profiling of individual cells [92]. The mRNAs of encapsulated cells are tagged onto unique barcoded beads and followed by RT-PCR. The platform was able to identify 39 cell populations from nearly 45,000 mouse retinal cells. With a similar approach to co-encapsulation of cells with barcoded beads, other platforms utilize different chemistry to achieve appropriate sequencing libraries for not only gene expression profiling, but also single nucleotide variation, copy number variation, and multi-omics analysis.

Aside from single cell transcriptomic analysis, the droplet microfluidic platform also has a significant contribution to the whole-genome analysis. However, single-cell whole genome analysis is challenging in sample preparation due to a minuscule amount of DNA from a single cell, fidelity of whole-genome amplification, verification of sequences used for variant calling, and bioinformatic analysis [94]. Generally, single-cell whole-genome analysis workflow is more complex as opposed to transcriptomic analysis, which includes releasing DNA from the nucleus, fragmentation, amplification, and tagging with unique molecular barcodes. To resolve this intricate workflow with a droplet microfluidic platform, droplet manipulation techniques and re-encapsulation are performed to carry out multiple biochemical reactions. Lan et al. developed the single-cell genomic sequencing (SiC-seq) that performs isolation, fragmentation, and barcoding of single-cell genome in droplets, followed by Illumina sequencing of pooled DNA [95]. SiC-seq approach can process over 50,000 cells per run with on-chip library preparation steps. The main strategy of SiC-seq involves single cell encapsulation in hydrogel droplets and solidified by cooling, which enables subsequent steps on cell lysis and protease treatment in a tube after droplet pooling. The isolated cells in hydrogel can circumvent any DNA crosstalk from neighboring cells. The cell-containing hydrogel beads are re-encapsulated in aqueous droplets along with tagmentation reagent for DNA fragmentation. Subsequently, a batch of droplets that contain PCR mix is passively merged with another batch of barcoded droplets through channel geometric constriction to form merged PCR-bar droplets. Lastly, the hydrogel droplets with tagmented genomes merge with PCR-bar droplets through the second merger, followed by thermal cycling for amplification. The resulting droplets are pooled together for the Illumina sequencing protocol. SiC-seq exploits the compartmentalization and high-throughput advantages from droplet microfluidics to enable on-chip library preparation, which reduces manual errors and increases the quality of the sequencing library. However, SiC-seq requires re-encapsulation, droplet manipulation, and multiple microfluidic chips, which increase workflow complexity and require exquisite control of fluid flows for droplet merging. Aside from the effect of biochemical reactions, it is important to note that the quality of the sequencing library is also highly susceptible to the process of re-encapsulation and serial droplet merging. In addition, the droplet reloading process requires delicate handling to ensure the integrity of droplets and the success of subsequent steps. Many existing commercialized droplet microfluidic platforms, such as 10x Chromium and Mission Bio Tapestri, require only manual cell loading into the microfluidic device without any intervention in between the library preparation process [96,97]. Thus, future development on simplifying the overall workflow and system automation will increase technology adoption.

In addition, single-cell copy number variation (CNV) has also been demonstrated using droplet microfluidic platform to reveal cellular genomic heterogeneity and clonal evolution [98]. Furthermore, 10x Genomics developed a workflow that involves single cell encapsulation along with agarose gel to form cell beads upon solidification [96]. After cell lysis and the washing step, the released genomic DNA remains inside the hydrogel cell beads, while digested proteins are removed. The cell beads are subsequently introduced into another droplet generation device to co-encapsulate with molecular barcoded beads and enzyme reagents for tagging genomic DNA. The resulting droplets are pooled together for off-chip PCR and sequencing to obtain CNV sequencing results. The single-cell CNV workflow from 10x Genomics requires fewer steps in comparison to SiC-seq, which is dictated by the number of chemical reactions for the library preparation.

This review only highlights a few applications on single-cell genomic analysis using droplet microfluidics, readers are recommended to these cited reviews for further detailed insights on single-cell omics analysis [28,99,100]. Taken together, droplet microfluidics plays a significant role in library preparation for single cell genomic analysis. The ability to compartmentalize individual cells and barcoded beads are the major advantages in droplet microfluidics to retain genomic information originating from the same cell within a micro-reactor, which is challenging if it was done in bulk solution. Furthermore, the nature of high-throughput processing capability in droplet microfluidics is also attractive for single-cell genomic analysis, since it can process tens of thousands of cells per run, which overcomes the physical constraints in microvalve and microwell platforms. Upon encapsulation of cells in droplets, a number of droplet manipulation techniques can also be used to perform serial complex biochemical reactions. As a result, droplet microfluidics technology is now used in many library preparation steps for single-cell analysis and adopted in several commercialized applications.

### 4.2. Proteomics

Integrating with fluorescence-activated droplet sorting (FADS), droplet microfluidics makes it possible to identify and isolate millions of antibody-secreting cells in an automated manner, whereas the micro-well and valve-based microsystem can only process hundreds or thousands of cells [101]. In addition, the droplet microfluidics further extends the breadth of capabilities in functional screening for antibody-secreting cells by not only detecting cell surface receptors but also cell-secreted molecules, such as cytokines [102]. Conventionally, a fluorescence-activated cell sorting (FACS) or flow cytometry that relies on the binding of cell surface markers is used to select cells of interest to acquire statistical information from a large population of cells. To select cells based on secreted molecules, additional analytical tools, such as the enzyme-linked immunosorbent assay (ELISA), enzyme-linked immunospot (ELISPOT), or intracellular cytokine staining, are required to further discriminate individual cells within a large population. However, these existing methods prevent high-throughput analysis and recovery of viable cells for subsequent clonal expansion. A droplet-based microsystem can circumvent these limitations by taking advantage of the confinement of a single cell in a highly miniaturized droplet and FADS. Upon compartmentalizing of a single cell with fluorescent detection antibodies or antigen-specific antibody functionalized microbeads in a droplet, each minuscule droplet serves as an independent microreactor that eliminates cross-contamination from neighboring cells. The drastic reduction of reaction volume in droplets allows the concentration of secreted molecules from an individual cell to reach a detectable signal, whereas bulk analysis would be incapable of achieving due to large background noise.

To exploit the advantages in screening and sorting from the droplet-based microfluidic system, El Debs et al. performed a functional screening and sorting of hybridoma cell clones producing 4E3 antibodies that inhibit congestive heart failure drug target angiotensin-converting enzyme (ACE-1) [103]. The heterogenous hybridoma cells were encapsulated along with recombinant ACE-1 enzyme into 660 pL droplets. The performance of cell encapsulation followed a Poisson distribution with 0.3 as the average number of cells per droplet. The resulting droplets were collected to incubate off-chip for 6 h to acquire a sufficient amount of secreted antibody concentrations (around 20 µg/mL). The droplets were re-introduced into another microfluidic chip for fusing with the second type of droplets containing the fluorogenic ACE-1 substrate at a rate of up to 50 Hz and over 99% efficiency. The droplet with low fluorescence intensity indicating ACE-1 inhibition were sorted at a rate of 5 × 10^4^ cells per hour or 14 droplets per second. The sorted 4E3 producing hybridoma cells were recovered for further off-chip characterization with up to 9400-fold enrichment. Since this approach is capable of screening up to 300,000 cells per run and does not require clonal expansion, it can be further extended to process non-immortalized primary B-cells. Nonetheless, some limitations still exist in the presented platform in terms of sorting throughput and system complexity. The sorting time would be significantly lengthened when screening millions of ASCs due to the low-throughput operation (14 droplets s^−1^) in FADS. The necessity of droplet fusion and sorting also increases the system complexity that hinders its wide adoption in the biological research field.

Thus, another screening method involving co-encapsulation of functionalized microbeads was developed to simplify the workflow while retaining the ability to detect secreted molecules. Konry et al. utilized avidin-coated microbeads (0.9 µm) that were conjugated with biotinylated IL-10 monoclonal capture antibodies as biosensors to detect the presence of IL-10 from T cells (Figure 9) [102]. These functionalized microbeads and CD4 + CD25 + regulatory T cells were co-encapsulated along with anti-human IL-10 FITC conjugated antibodies in ~1.8 nL droplets. Thanks to the restricted droplet volume, the secreted antibodies released by a single cell can quickly reach detectable concentrations. With only two hours of on-chip incubation, the secreted IL-10 antibodies were bound to the microbead, followed by the localization of fluorescent secondary antibodies around the bead surface. The localization of fluorophores of the secondary antibodies resulted in a narrow, high-intensity fluorescence peak, whereas a broad, low-intensity fluorescence peak could be observed for droplets with an absence of IL-10 because of dispersion over the entire droplet. This one-step approach to detect single-cell secretion eliminated the need for droplet fusion and numerous washing steps required by other approaches.

This detection technique was also used by Mazutis et al. to develop a binding assay for screening IgG antibodies secreted from a single mouse hybridoma cell [104]. A heterogeneous mixture of antibody-producing and non-producing hybridoma cells were co-encapsulated with anti-mouse-Fc capture antibodies conjugated streptavidin beads and fluorescently labeled secondary antibodies into 50 pL droplets. After 15 min of off-chip incubation, the resulting droplets were re-injected into a second microfluidic device for sorting ASCs using FADS. The binding of secreted antibodies to functionalized microbeads with fluorescent probe generated a distinguishable fluorescent signal, which enabled droplet sorting at 200 Hz. The coupling of droplet microfluidic technology and microbead biosensor demonstrated a versatile approach to screen cells based on cell-surface binding or secretion molecules in high throughput.

However, one inevitable problem in a strong variation of fluorescence signal persists due to the different position of cells or beads with bound fluorescently labeled secondary antibodies in droplets, either closer to or further away from the focal plane and the center of the laser spot. Shembekar et al. has resolved this limitation by employing a dual-color normalized fluorescence readout to detect antibody binding in a quantitative manner [105]. The dual-color normalization technique was applied to antibody binding assay, involving co-encapsulation of a hybridoma (OKT 9 cells secrete antibodies against the transferrin receptors and H25B10 cells secret non-related antibodies) and target tumor cells (K562 cells) in a single droplet. The OKT 9 hybridoma cells were successfully enriched 220-fold after sorting 80,000 clones at a rate of up to 40 Hz. Moreover, this technique demonstrated a high sensitivity of quantifying as little as 33 fg of antibody. This platform not only enables quantitative screening of ASCs for antibody expression but also specific antibody binding without focusing. Without the need for cell proliferation, this platform is also compatible with processing non-immortalized cells, such as primary plasma cells. Overall, the presented droplet microfluidic systems enable efficient screening of ASCs and provide flexibility to screen based on surface-binding or antibody secretion. This technology would pave the way for therapeutic antibody discovery or research purposes.

In addition to the screening of ASCs, the primary DNA sequence of immunoglobulin repertories is the key to understanding the adaptive immune response and diversity of the antibody repertoires for therapeutic antibody discovery. A unique advantage of droplet microfluidics when compared to other screening techniques is the ability to link genotype with phenotype. After sorting of positive antibody-secreting cells, DNA barcoding technology can be incorporated by tagging individual cells with a barcoded bead containing unique oligonucleotides to reveal a sequence of immunoglobulin repertoires using next-generation sequencing. The barcoded beads can be introduced by re-encapsulating barcoded beads with cells in droplets or utilizing on-chip manipulation for droplet fusion. Existing techniques for high-throughput immune repertoire sequencing have low cell throughput (10^4^–10^5^ cells), which is far fewer than the 0.7 × 10^6^–4 × 10^6^ B cells contained in a typical 10 mL blood sample [106]. Such a low sample size cannot provide sufficient depth of analysis regarding the human paired heavy- and light-chain (V_H_-V_L_ pairs) of antibodies. To address the current limitations, DeKosky et al. co-encapsulated a single B cell with magnetic mRNA capturing beads into droplets containing lysis buffer, with subsequent use of RT-PCR to generate V_H_-V_L_ amplicons for next-generation sequencing [107]. This method made ultra-high-throughput V_H_-V_L_ repertoire sequencing possible and offered new immunological insights on the identity, frequency, and pairing propensity of shared genes, the detection of allelic inclusion in healthy individuals, and the occurrence of antibodies with features with broadly neutralizing antibodies to rapidly evolving viruses such as HIV-1 and influenza. However, the sequencing results represented a collective of V_H_-V_L_ repertoires after pooling the captured mRNAs that missed the individual V_H_-V_L_ pairing information at a single-cell level.

Recently, Gérard et al. developed the CelliGo system that combines high-throughput screening for IgG activity with the sequencing of paired antibody V_H_-V_L_ genes in single-cell resolution [108]. The mouse splenocytes were co-encapsulated with bioassay reagents in 40 pL droplets at a rate of 2100 droplets per second for screening IgG secretion. Upon in-tube incubation for the collected droplets, fluorescent activated droplet sorting was employed to select positive IgG-secreting cells at roughly 600 droplets per second. The sorted cells were recovered by emulsion breakage and re-encapsulated with molecular barcoded hydrogel beads for heavy and light chain domain (V_H_ and V_L_) primers, lysis buffer, and reverse transcriptase for generating the sequencing library. The reverse-transcribed cDNAs were obtained by emulsion breakage and further amplified for Illumina pair-end sequencing. Owing to the unique barcode and elimination of cross-contamination among droplets, a native cognate pair of V_H_ and V_L_ information was preserved and identified by tracing the sequencing results that shared the same barcode. The CelliGO platform offers the capacity to obtain IgG repertoire by high-throughput screening of millions of cells on both soluble and membrane-bound target for a reduced processing time (20 days from cell harvesting to next-generation sequencing). The success of bridging phenotype to genotype for repertoire characterization demonstrates tremendous potential in antibody discovery for unique functional activities for therapeutic purposes.

Notwithstanding the unique enabling techniques brought by droplet microfluidics, it still has a few noteworthy limitations. The co-encapsulation efficiency of cell-cell or cell-bead in a droplet is governed by double Poisson statistics, which is limited to only 13% of effective droplets. In actual practice, however, the cell encapsulation is performed such that the majority of droplets should contain no more than one cell of the same type, whereas multiple beads per droplet can be tolerated depending on the type of subsequent analysis. Thus, a diluted cell suspension is often required to minimize encapsulation of two or more cells of the same type, which further reduces the co-encapsulation efficiency. Although alternative encapsulation techniques, including Dean-flow assisted ordering and inertial ordering, were developed to address the low co-encapsulation efficiency, these techniques are restricted to homogenous cell mixtures and dependent on the physical properties of the particles [109,110]. The ability to efficiently co-encapsulate cell-cell or cell-bead is of increasing importance in high-throughput single-cell analysis. Due to Poisson statistic limitation, droplet sorting is often necessary to eliminate the large fraction of non-functional droplets, which presents another challenge in system complexity. The process of droplet re-injection and sorting would require hands-on expertise that may hinder its adaptation to other research fields. Moreover, the throughput of droplet sorting typically ranges from 50 to 600 droplets per second, which becomes the bottleneck in throughput and increases the workflow runtime. Thus, the development of the droplet sorting technique is under active research to further improve the sorting rate without compromising accuracy [111].

Alternatively, droplet microfluidics can also be utilized to generate hydrogel microparticles for compartmentalization of cells with a precursor (an aqueous monomer) solution in an immiscible nonpolar solution followed by polymerization. The permeability of hydrogel is a great candidate to serve as a miniaturized cell culture chamber that enables the exchange of reagents. Akbari et al. exploited this property to efficiently screen for anti-TNF-α secreting cells from a heterogeneous mixture of antibody-secreting cells [112]. Specifically, a significant advantage of utilizing hydrogels as opposed to water-in-oil emulsions is the nanoporous structure from the alginate microparticles. These nanopores allow bidirectional diffusion of small molecules to remove any unbound fluorescent probes through the washing step, while the large molecules and cells are trapped inside the alginate microparticles. Thus, after several washing steps, the homogenous fluorescent distribution of microparticles revealed the efficient capturing of secreted antibodies without observing crosstalk between neighboring microparticles and high background signal noise. The capability of reagent exchange for the encapsulated cells permits more complex cellular analysis that requires multiple washing steps. This technique would be beneficial for applications in single cell/molecule analysis, such as single-cell culture, sequencing, and molecular evolution [113].

Furthermore, functionalized hydrogel microparticles can also be used for microbead-based immunoassays. Although the microbead-based immunoassays offer enhanced assay flexibility, multiplexing, and improved sensitivity over ELISA, the main challenge is in the immobilization of antibodies onto the microbead surfaces, which is prone to damage an antibody’s three-dimensional molecular structure. To address this issue, Chen et al. developed a one-step droplet-based microfluidic method to produce nanoporous alginate microparticles with immobilization of different antibodies for specific binding and binding affinity tests [114]. The sodium-alginate microparticles were generated with a diameter ranging from 40 to 60 µm followed by submerging the collected microparticles in an external buffer solution containing calcium ions and antibodies (anti-BCG IgY or anti-E. coli IgG). The microparticles were crosslinked, whereas the antibodies were immobilized onto the microparticle surfaces during the gelation process. Owing to the high surface area-to-volume ratio of the functionalized alginate microparticles, the increased probability of binding target bacteria and the immobilized antibodies further improves the detection limit. Lastly, the functionalized microparticles also demonstrated a comparable binding affinity with the ELISA approach. The functionalized alginate microparticles have the potentials to include other biological, magnetic, and electrical properties in the droplet microfluidic platform, which offers more accurate and complex characterization studies of ASCs.

To conclude, a droplet-based microfluidic system not only offers phenotypic high-throughput single-cell screening based on a variety of bioassays, but also enables the study of cellular diversity in linking genotype and phenotype. The capacity to perform bioassays in single-cell resolution enables unprecedented throughput and eliminates the requirement for cell immortalization. The discussed platforms in the single-cell genomic and proteomic analysis only represent a subset of droplet-based applications [51]. As the droplet microfluidic technologies continue to develop, many new analytic capabilities will emerge to advance the workflow of sequencing library preparation, antibody discovery, and research at a lower cost with increased efficiency and breadth of biological applications.

## 5. Conclusions and Outlooks

Microfluidic systems offer unprecedented opportunities in studying multiplexed, high-throughput, single-cell screening and profiling of millions of single cells, and yet enabling characterization of genotype to phenotype among a vast diversity of cells. The manipulation of fluids in micro-scale is the essence of microfluidics, which makes isolating and compartmentalizing single cells in pico- to nano-liter compartments possible and further enhances the sensitivity by increasing analyte effective concentration in small confinement.

The three main microfluidic compartmentalization platforms, namely, microvalves, microwells, or droplets, can significantly increase the processing throughput, increase detection sensitivity, and reduce labor error and costs in comparison to the conventional microtiter plate method. Specifically, these three compartmentalization systems discussed here have respective unique advantages that make each system preferentially favorable for different types of single-cell analysis. The microvalve-based system offers precise manipulation of cells and reagents for a more complex analysis in an automated manner, which enables them to be suitable for applications where meticulous transport of cells or reagents is critical. The microwell-based system provides simplicity to the workflow for cell loading and defined spatial locations for retrieval at an intermediate throughput. Lastly, the droplet-based system also enables precise control of liquid handling but also allows high-throughput processing capability that is suitable for screening millions of cells. A detailed comparison of the aforementioned microfluidic compartmentalization platforms is illustrated in Table 1.

In summary, microfluidic systems facilitate the study of single cells under different conditions at genomic and phenotypic levels. Given the unique benefits and limitations offered by different microfluidic platforms, there are no general metrics to unambiguously conclude that a specific microfluidic platform is more competitive than the other in single-cell analysis. When selecting an appropriate microfluidic device for single-cell analysis, researchers should carefully consider the distinctive advantages offered by each microfluidic platform, complexity of fabrication, throughput, and type of biological assay to be incorporated. The characteristics of a particular microfluidic compartmentalization platform and the objectives of single-cell analysis will affect the incorporation of existing assays into the microfluidic device and future efforts. Collectively, the microfluidic single-cell compartmentalization platform is already playing a critical role in a new era of revealing heterogeneity and treating rare diseases with a comprehensive single-cell profile. Aside from the aforementioned microfluidic systems, it is also worth mentioning that paper microfluidics has also attracted increasing interest for single-cell analysis due to its low cost, portability, and facile integration with other devices [9].

The continued development of microfluidic compartmentalization platforms for single-cell analysis will accelerate the process of antibody discovery, drug screening, and improve our understanding of genomics and transcriptomics of single cells. Despite previous endeavors, there are still many unsolved challenges in the field of microfluidic systems, such as sample evaporation, quality control, system complexity, adoption in clinical and industry settings, etc. However, as more and more microfluidic techniques are developed, many novel and multi-analytical capabilities for profiling single cells are on the horizon and expected to emerge. By gathering additional information from cells, such as proteome, transcriptome, and epigenome, the microfluidic compartmentalization platforms hold promise for leading to a more comprehensive study of single-cell analysis, complex immune responses, and the possibility of developing personalized medicines [117,118,119]. Therefore, microfluidic systems have demonstrated their significant impact in the fields of genomics, proteomics, antibody discovery, and biomedicine and are primed to become a standardized tool as the technology maturates.

## Figures and Tables

**Figure 1 biosensors-12-00058-f001:**
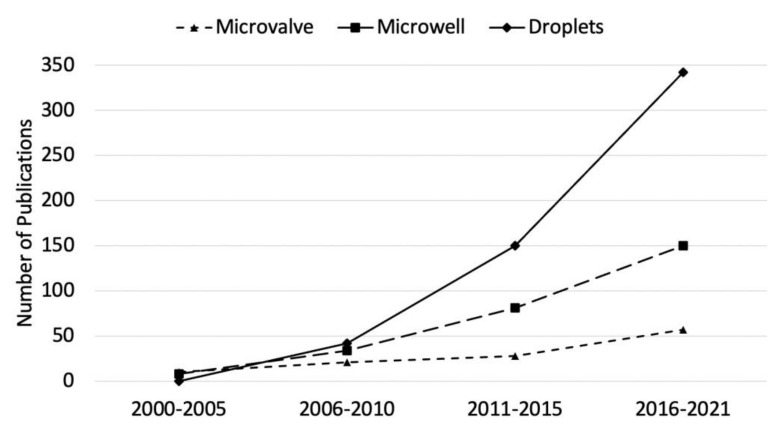
The number of publications from Web of Science from the year 2000 to 2021 on microvalve, microwell, and droplet microfluidics for single-cell analysis.

**Figure 2 biosensors-12-00058-f002:**
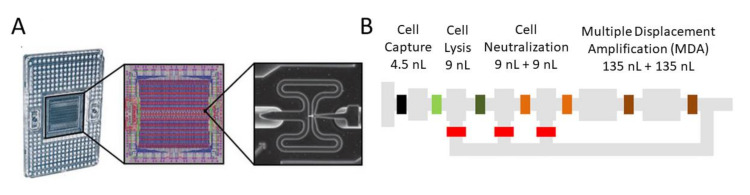
Fluidigm C1 IFC Autoprep System. (**A**) Illustration of C1 IFC: (**left**) IFC with carrier; (**middle**) diagram of the IFC connection; (**right**) example of one single cell immobilized in one captured site on IFC. (**B**) Schematic of the C1 IFC operation. Reprinted with permission.

**Figure 3 biosensors-12-00058-f003:**
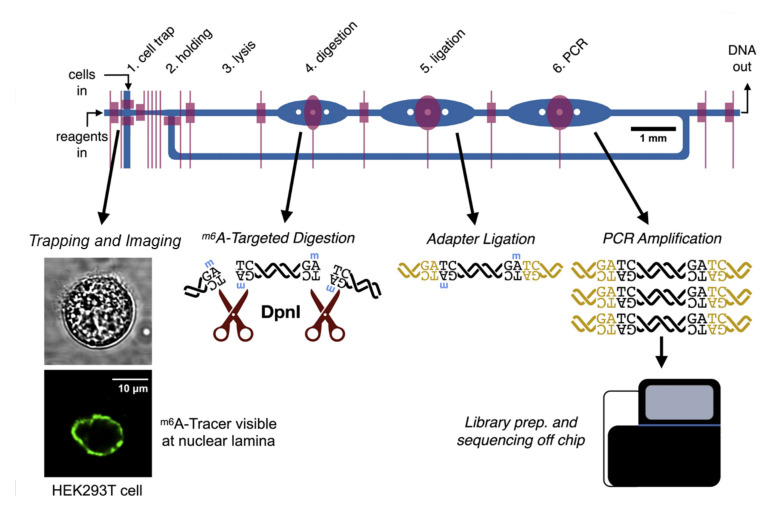
Schematic of DamID protocol and the function of each chamber [40]. Reprinted with permission.

**Figure 4 biosensors-12-00058-f004:**
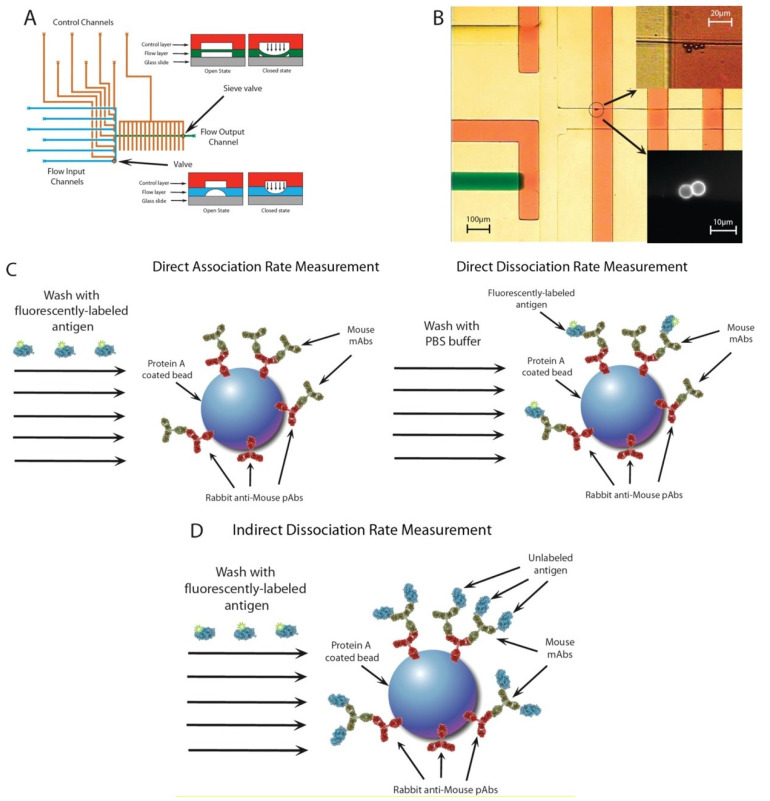
Illustration of microfluidic device and the association/dissociation kinetics. (**A**) The configuration of the device. (**B**) Visualization of the inlets (yellow and green) and the control channels (red). The image at the bottom right shows the beads captured by the sieve valves. (**C**) Measurement of association rate. (**D**) Measurement of dissociation rate [48]. Reprinted with permission.

**Figure 5 biosensors-12-00058-f005:**
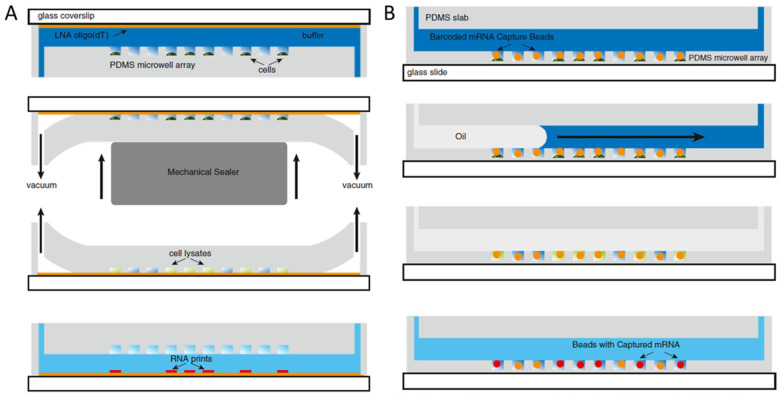
Schematic of (**A**) scRNA printed on a glass slide and (**B**) captured on a barcoded bead [68]. Reprinted with permission.

**Figure 6 biosensors-12-00058-f006:**
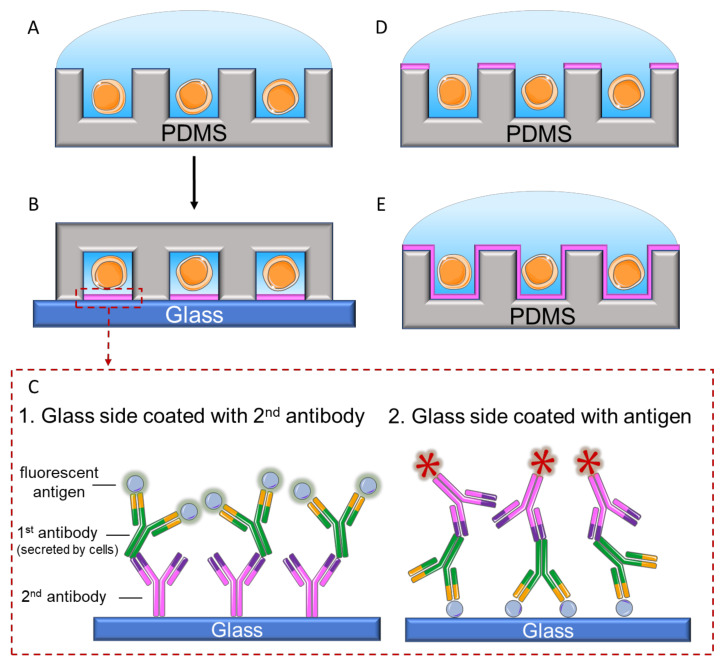
Techniques of capturing the antibodies secreted by single cells in microwells. (**A**) single cells are captured in microwells. (**B**) Microwells are sealed with 2nd antibody-coated or antigen-coated glass slide. (**C**) The presence of the target antibody is visualized by fluorescently labeled antigen or secondary antibody. (**D**,**E**) Functionalization of the well. (**D**) Functionalize the top surface of the well, as reported by Muraguchi et. al. [54,59]. (**E**) Functionalize the top and the inner surface of the well, as studied by Torres et al. [79].

**Figure 7 biosensors-12-00058-f007:**
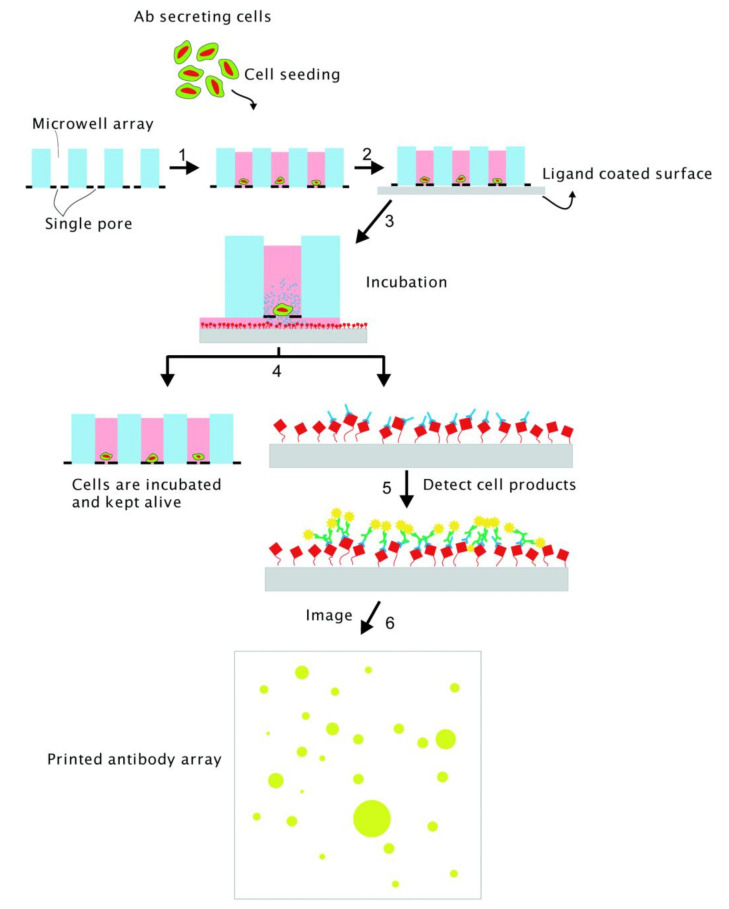
Illustration of antibody-secreting cells using VyCAP. The microwells have a 5 µm hole at the bottom. The wells are brought in contact with a ligand-coated membrane (gray) and the antibodies were captured on the membrane. The captured antibodies were visualized by fluorescent immunostaining [83]. Reprinted with permission.

**Figure 8 biosensors-12-00058-f008:**
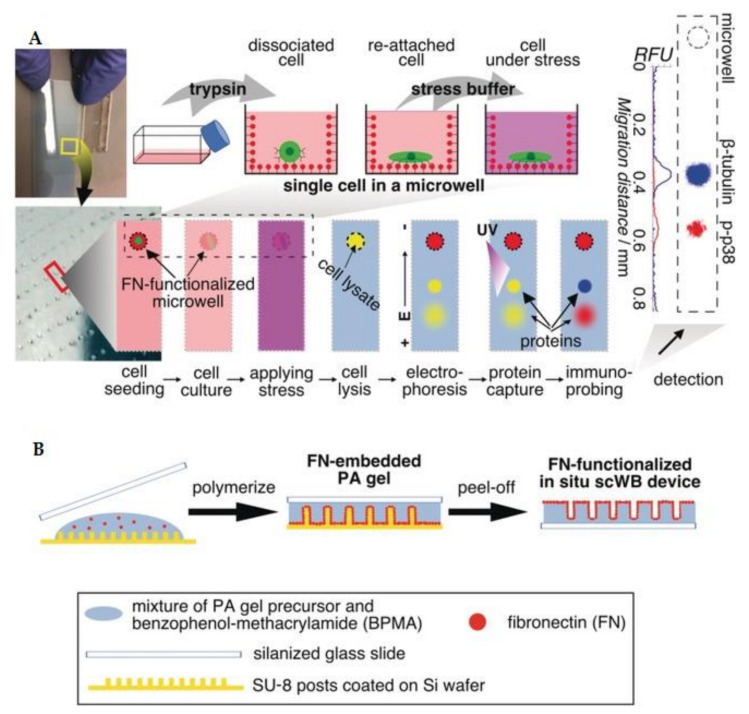
In situ single-cell Western blot (in situ scWB). (**A**) Workflow of in situ scWB illustrating an array from one of the 2000 arrays on a device. (**B**) Preparation of FN-functionalized PA microwell [85]. Reprinted with permission.

**Figure 9 biosensors-12-00058-f009:**
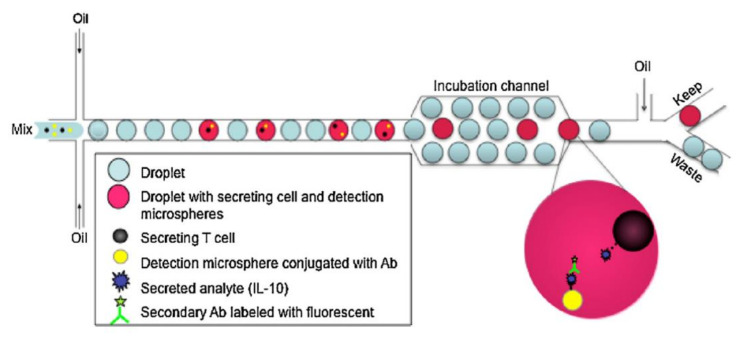
Schematic illustration of the configuration of droplet based microfluidic platform. The center stream contains a mixture of microspheres, cells, and secondary antibodies (mix), while the two opposing side streams contain the oil phase. The encapsulated cells proceed into the downstream incubation region for cell analysis and can then be sorted based on secretion of the interrogated analyte [102]. Reprinted with permission.

**Table 1 biosensors-12-00058-t001:** Comparison of microfluidic compartmentalization platforms.

	Microvalves	Microwells	Droplets
Sample Volume	2–700 nL/chamber [27,44]	100 pL–1.5 nL/well [68,82]	1 fL–10 nL/droplet [19,89]
Multiplexing Capability	High	Low	Medium
Selective Cell Retrieval	Easy	Medium	Difficult
Throughput	10–300 cells/run [24,115,116]	100–10^5^ cells/run [66,88]	>10,000 cells/run [92]
Scale-up Capacity	Low	Medium	High
Ease-of-Operation	Difficult	Easy	Medium
Integration Potential	High	Low	Medium

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
