# Peer review of "Microfluidic Compartmentalization Platforms for Single Cell Analysis"

_biosensors, 2022, doi:10.3390/bios12020058_

Round 1

Reviewer 1 Report

The paper presents a review of recents achievements of microfluidic devices for single cell genomic and proteomic analysis. The authors focus on three main branches of the devices: valves based, micro and nanowells, and droplets. All of these techniques provide powerful tools for single cell analysis and are well described in the paper. In the conclusion there is a comparison of these methods indicating their strong and weak sides. The language and the presentation is good however there are several issues, therefor proofreading is required. 

Comments

1. correct the citation of the references in the main text. In a lot of places there is a phrase "Error! Reference source not found"

2. in the multiwell devices there is a slightly different concept based on hydrodynamic traps. Logically they are very similar but may provide some advantages. The reviewer recommend to briefly mention this type of devices and summarize their weak and strong sides.

3. The reviewer recommends to add some historical notes of evolution of microfluidic devices for single cell analysis. According to the references publishing years first microvalve and microwell devices were introduced for single cell analysis. But now it seems that droplets displace them in the considered applications. Its worth adding the graph illustrating the number of papers made using these techniques to show their demand.

4. Because of enormous growth of droplet microfluidics are their any specific tasks where microvalves/microwells still have advantage? It the conclusions section the reviewer recommends to add some discussion about future promising ways of using and development of all these technologies and their demand for single cell analysis.

Author Response

Thank you for your positive comments and constructive advises. The citations in this review paper have been corrected. Under the microwell section, we’ve also added the component that discusses the single-cell analysis on hydrodynamic traps. The historical notes on microfluidics devices based on these three branches have been added in the introduction section with the number of publications according to Web of Science. Given the three microfluidic platforms, each consists of unique advantages and disadvantages in comparison to other platforms. There is no definitive metric that could conclude the best platform for single-cell analysis. The choice of microfluidic platform should depend on type of single-cell analysis and type of biological assays. Further clarification is added to the conclusion section. We hope you will find this revision is in accordance with the scope of the Biosensors’ special issue on “Microfluidic Systems and Its Applications in Single-cell Analysis”. Thank you.

Reviewer 2 Report

The authors presented a literature review on microfluidic platforms exploiting compartmentalization for single cell analysis. Overall, the review appears adequate and interesting to the reader. The authors successfully presented the state of the art, while also addressing advantages and limitations of each approach. Overall, I recommend this paper for publication, after some minor remarks are addressed:

  • Cross-references to images within text are not working properly.
  • Line 214, reference missing.
  • Table 1 should be more quantitative, as it fails to convey meaningful information to the reader in its current form.

Author Response

Thank you for your positive comments and constructive advises. References have been reviewed and corrected. Regarding Table 1 that compares and contrasts among the three microfluidic platforms, we could use engineering table to give a score to each category instead of using “high, medium, or low”. We tried to use more quantitative for each box as much as possible. However, some categories seem to be most appropriate to use more qualitative comparison, such as high, medium, or low, as we are comparing relative to one another. We hope you will find this revision is in accordance with the scope of the Biosensors’ special issue on “Microfluidic Systems and Its Applications in Single-cell Analysis”. Thank you.

Reviewer 3 Report

Manuscript Tile.  Microfluidic Compartmentalization Platforms for Single Cell 2

Analysis

Journal: Biosensors

The manuscript entitled “Microfluidic Compartmentalization Platforms for Single Cell Analysis” describes single-cell microfluidics. The article is timely and highlights important information about microfluidic systems, and is of potential interest to the readers of “Biosensors”. Before I recommend publication, the manuscript needs to be improved further.

Authors should highlight the rationale of the study in introduction section. Please clarify why there is a still need to write a review on this topic? Authors are suggested to read some related reviews and structure introduction in light of previous reports: Trends in Analytical Chemistry 143 (2021) 11641 https://doi.org/10.1016/j.trac.2021.116411; Micromachines 2019, 10(2), 104; https://doi.org/10.3390/mi10020104 ; Trends in Analytical Chemistry 117 (2019) 13e26 https://doi.org/10.1016/j.trac.2019.05.010; Analyst, 2019,144, 766-781 https://doi.org/10.1039/C8AN01186A etc.

  1. Authors cited only three papers in introduction. Many statements need citations (implies to the whole manuscript). For example, “A number of single cell analysis methods have been developed…….” needs citation. “10x Genomics developed a workflow that involves single cell encapsulation along with agarose gel to form cell beads upon solidification. After cell lysis and washing step, the released genomic DNA remains inside of a hydrogel cell beads, while digested protein are removed” no citation is given.
  2. It is not clear whether figures are reproduced with permission. Authors should add a copyright statement clarifying that the figures are reprinted or reproduced with permission, and may also add the license number.
  3. Explain all acronyms at first use.
  4. Line# 569-571, author stated that “Future development on simplifying overall workflow and system automation will further increase technology adoption”. Explain the statement with some examples.
  5. Line# 588 “This review only highlights a few applications on single-cell genomic analysis using droplet microfluidics.” It is suggested that authors recommend a related article (review) for further insights on single-cell genomic analysis. Please remember that each section may appeal different readers. It will help readers to explore relevant literature.
  6. Paper microfluidics is an emerging field and has shown significant progress over the last two decades. It is recommended that authors add a section to discuss paper microfluidics for single-cell analysis.
  7. Conclusions and outlooks: Add some perspectives in this section.
  8. Sample evaporation may be a persistent challenge in microfluidic systems. How can we solve this issue? This is especially true when using volatile or low-boiling compounds to screen libraries.
  9. Please carefully check the typos and errors throughout the manuscript. For instance, line # 28 “These population-level cellular analysis plays…. à These population-level cellular analyses play…., line # 96 “singles cells”, line # 142 “which is has not..”

Author Response

Thank you for your positive comments and constructive advises. Introduction has been revised according to the given suggestion on the introduction section. We’ve also added appropriate and missing citations throughout the manuscript. We appreciate your insights on the emerging field in paper microfluidics. We’ve also realized the rising interest in paper microfluidics; however, we feel that it is outside the scope of our review since we focus on compartmentalization of single cells and its related applications. This has been briefly discussed in conclusion and cited related review papers for readers who are interested in paper microfluidics. Conclusion is also revised to be more comprehensive in summarizing the three major microfluidic platforms. We hope you will find this revision is in accordance with the scope of the Biosensors’ special issue on “Microfluidic Systems and Its Applications in Single-cell Analysis”. Thank you.
